# Beliefs and Experiences of Individuals Following a Zero-Carb Diet

**DOI:** 10.3390/bs11120161

**Published:** 2021-11-23

**Authors:** Cleo Protogerou, Frédéric Leroy, Martin S. Hagger

**Affiliations:** 1Psychological Sciences and Health Sciences Research Institute (HSRI), University of California, Merced, 5200 N. Lake Road, Merced, CA 95343, USA; 2Department of Psychology, University of Cape Town, Rondebosch, Cape Town 7701, South Africa; 3Research Group of Industrial Microbiology and Food Biotechnology (IMDO), Department of Bioengineering Sciences, Faculty of Sciences and Bioengineering Sciences, Vrije Universiteit Brussel, Pleinlaan 2, B-1050 Brussels, Belgium; frederic.leroy@vub.be; 4Faculty of Sport and Health Sciences, University of Jyväskylä, Seminaarinkatu 15, FI-40014 Jyväskylä, Finland

**Keywords:** carbohydrate restriction, ketogenic-type diets, wellbeing, quality of life, thematic analysis, lived experience

## Abstract

The adoption of carbohydrate-restrictive diets to improve health is increasing in popularity, but there is a dearth of research on individuals who choose to severely restrict or entirely exclude carbohydrates. The present study investigated the beliefs and experiences of individuals following a diet that severely limits, or entirely excludes, dietary carbohydrates, colloquially known as a ‘zero-carb’ diet, for at least 6 months. Zero-carb dieters (*n* = 170) recruited via a social networking site completed an online qualitative survey prompting them to discuss their motives, rationale, and experiences of following a low-carb diet. Transcripts of participants’ responses were analyzed using inductive thematic analysis. Results revealed that participants’ decision to follow a zero-carb diet was driven by health concerns and benefits. Participants expressed a strong social identity and belongingness to online zero-carb communities. Participants reported strong intentions to follow the diet indefinitely. Shortcomings of the diet centered on experienced stigma; lack of support from healthcare providers and significant others; limited access to, and high cost of, foods; and limited scientific data on the diet. Further research into the benefits and shortcomings of zero-carb diets across settings and populations is warranted, and guidelines for healthcare professionals on how to support individuals following a zero-carb diet are needed.

## 1. Introduction

National dietary guidelines generally recommend a reduction in fat and red meat consumption and an increase in carbohydrate consumption to promote good health [1,2]. These recommendations have been based on theoretical links between fat consumption and coronary heart disease risk [3], and associations between red and processed meat consumption and incidence of cardiovascular disease and certain types of cancer [4]. Despite these recommendations, diet-related morbidity as well as cardiovascular disease, diabetes, and obesity and overweight, have risen markedly, while fat and red meat consumption has declined and carbohydrate consumption has increased [5,6]. In parallel, a substantial body of evidence has emerged linking high carbohydrate consumption, particularly dietary sugars, to increased risk of chronic conditions including metabolic syndrome, obesity, diabetes, and related chronic illnesses [7]. Other research claims that the relations between red (including processed) meat consumption and increased risk of chronic disease are based on low-quality evidence [8]; and that meat consumption provides a plethora of health benefits [9,10].

Against this backdrop of inconsistent evidence and ongoing debates on the harms and benefits of animal-sourced food and fat consumption [11], some populations have been reducing their carbohydrate consumption for health reasons, leading to the popularity of low-carbohydrate and ketogenic-type diets [12]. There is debate over the health benefits and risks of low-carbohydrate and ketogenic diets. Some evidence argues against these diets, putting forth short-term (e.g., constipation, headache, halitosis, muscle cramps, bloating, diarrhea) and long-term (e.g., decreased bone mineral density, nephrolithiasis, cardiomyopathy, anemia, and neuropathy of the optic nerve) health concerns [13,14]. However, following a carbohydrate restricted dies has also been linked to adaptive health outcomes, such as sustained weight loss, improved physique, reduced hunger, improved mood and cognition, better digestion, improved biomarkers, and higher quality of life [15,16]. In addition, a growing number of case reports suggest that carbohydrate-restrictive diets consisting almost exclusively of animal-sourced foods are effective in managing and even reversing chronic health conditions, including obesity, type 1 and 2 diabetes, Crohn’s disease, and epilepsy [17]. The underlying physiological mechanism hypothesized to explain the health-related benefits of carbohydrate reduction is nutritional ketosis [18]. Furthermore, the avoidance of plant antinutrients, such as phytates, lectins, oxalates, and fiber, has also been proposed as a factor explaining health benefits experienced through carbohydrate restriction [19]. The lack of consensus on the health benefits and risks of carbohydrate reduction could be explained, at least in part, by the absence of a singular low-carbohydrate diet [20]. Many types of low-carbohydrate and ketogenic-type diets appear in the published literature and in real-life, with variable adherence patterns, which hinders efforts to reach overarching conclusions on the health-related consequences of the diets.

Similarly, there are numerous versions of diets that severely limit or entirely exclude carbohydrate intake, colloquially labeled zero-carb diets. People eating zero-carb diets are known to consume, almost exclusively, animal-sourced foods (i.e., all types of meats and fish, eggs, and dairy), at the expense of plant-based foods (i.e., fruits, vegetables, legumes, and grains). These diets may have emerged in response to calls for “personalized” approaches to nutrition [21]. The prominence of the internet as a primary venue of health-related information and social interaction has also likely facilitated the emergence of zero-carb diets. There are many long-established virtual zero-carb communities with large followings. For example, the zero-carb subreddit was created in 2010 and currently counts 114,000 registered members (see https://www.reddit.com/r/zerocarb/, accessed on 13 April 2019). However, there has been, to date, no published empirical research on the factors that influence people to take up, and remain committed to, a zero-carb diet [22]. There is also no empirical research on the shared definitions and features of a zero-carb diet among those who follow them, which is important in order to document typical practices, and inform research on potential health benefits and risks. The rise in popularity of zero-carb diets in media outlets has created the substantive need for information for those already following these diets and those interested in taking up the diet in the future. Furthermore, the need for research in this area is important given the increasing evidence that carbohydrate-restrictive diets have beneficial health effects. The absence of research on zero-carb diets has also presented challenges to health-care professionals (e.g., physicians, nutritionists, psychologists) interested in establishing the factors that determine the diet uptake, and provide appropriate support for people following these types of diet.

The current study aimed to fill this evidence gap by exploring the beliefs and experiences of adults on zero-carb diets. Specifically, we aimed to identify the shared knowledge and understanding of what characterizes a zero-carb diet and to identify the sets of beliefs that underpin decisions to begin and maintain a zero-carb diet. Given the dearth of empirical research on zero-carb diets, we used a qualitative survey to gain insight into the beliefs and experiences of those eating zero-carb diets. Online surveys have been gaining traction as a qualitative research method to provide in-depth information on beliefs and behaviors [23]. The method offers numerous advantages for researchers, such as a “wide-angle lens” approach on the topic under investigation that allows the collection of diverse views, perspectives, and experiences of geographically dispersed populations; the elicitation of “within-group” views from marginalized and/or overlooked populations; heterogeneity, instead of typicality of views, when larger samples are recruited; and the quick collection of large data sets. The method also offers some advantages over traditional interview approaches by providing anonymity for respondents and greater control over when, where, and how they express their views, without feeling influenced or led by an interviewer or facilitator. For an overview of the online survey method, see Braun, Clarke, Boulton, Davey, and McEvoy [24].

## 2. Materials and Methods

### 2.1. Participants and Recruitment

Participants (*n* = 170) were recruited from Twitter, an online social media and micro-blogging site. Twitter has become a popular platform for social research, as it offers recruitment advantages including reduced costs and access to specific, large, hard-to-reach populations that are geographically and culturally diverse [25]. A tweet is a message posted on the platform and can act as an online advertisement of a study. When a study advertisement is retweeted (shared) with the online community mentioning the target population, it reaches the target population [26]. The use of personal Twitter accounts that are active and have many followers are seen as trustworthy by potential research participants, and likely to result in higher recruitment and participation rates [25,26]. At the time of data collection, the three authors’ Twitter accounts were active daily and totaled over 7000 followers. Data were collected by the first author and data collection lasted 10 days. To be included, participants had to be adults (≥18 years) following a zero-carb diet for at least six months prior to the date of data collection. This 6-month cut-off was set to exclude people that may have been following the diet capriciously and/or without substantial personal investment.

### 2.2. Design and Procedure

The current study was a qualitative survey, developed with Qualtrics, a web-based survey tool. The study advertisement was tweeted by the first author and subsequently re-tweeted by the two co-authors. The study advertisement was also pinned (prominently displayed) on the first author’s Twitter account during the recruitment period. The advertisement invited adults who had been following a zero-carb diet for at least 6 months to participate in a study “on zero-carb eating”. The invitation asked potential participants to contact the first author (through Twitter’s messaging function) to learn more about the opportunity to participate in the study. Interested participants who contacted the first author were provided an anonymized link that offered access to the online survey. Ethical approval of the study was granted by the Ethics Review Committee of the [University omitted for masked review] in April 2019 (reference number: PSY2019-012). We disclose that the first author’s interest in this topic was prompted by their own experiences with a zero-carb diet. Study design development was guided by an early version of Protogerou and Hagger’s [27] quality criteria for survey studies, and Braun et al.’s [24] frameworks for online survey research. Study materials and data are available online in the Appendix A.

### 2.3. Measures

The online survey comprised three sections. The first section provided participants with an information sheet and an informed consent form, followed by questions capturing participants’ demographic details: age, gender, country of residence, highest education level, occupation, mental health condition diagnosis, and disordered eating status. Disordered eating was measured using the Eating Disorder Screen for Primary Care (ESP) [28], an established screening tool that has been used across settings and populations [29]. The ESP comprises four questions (e.g., “Are you satisfied with your eating patterns?”) measured by a binary yes/no response scale. More than two “abnormal” responses, a “no” response to the first item and a “yes” response to others, are considered as a positive screen for an eating disorder.

The second section included open-ended questions focused on eliciting participants’ beliefs about their zero-carb diets. Questions were drawn and adapted as needed from established evidence-based instruments [30,31]. This level of specificity may, at first sight, appear at odds with traditional qualitative approaches, but the use of tested, clear, and non-ambiguous items is a recommended practice in online qualitative surveys, given that participants are not able to ask clarifying questions and need to hold a reasonable level of literacy [24]. The questions asked respondents to provide, in relation to eating a zero-carb diet, perceived advantages and disadvantages; approval and disapproval from significant others, such as family, peers, organizations; and the facilitating/inhibiting factors. Participants’ stated intentions to continue eating a zero-carb diet in the future were also recorded.

The third section was an open-ended question encouraging participants to provide additional comments and thoughts on zero-carb eating. Upon completion, participants were directed to a debriefing page. The full survey is available in the Appendix A.

### 2.4. Data Analyses

Participants’ responses to scaled survey items were summarized using means, standard deviations, and percentages. Participants answered open-ended questions in their own words and in as much depth as they chose. Responses to open-ended questions were high in volume and density; for example, participants’ responses to each open-ended question resulted in a minimum of 25 single-spaced pages (>10,000 words) of data. Data from open-ended questions were used verbatim to preserve participants’ personalized expression style. Responses to the open-ended beliefs items were thematically analyzed following processes described by Elo and Kyngäs [32] and Braun and Clarke [33]. Thematic analyses are systematic ways of analyzing data aimed at producing a “…condensed and broad description of the phenomenon, and the outcome of the analysis is concepts or categories describing the phenomenon” [32] (p. 108). In the current study, this was focused on identifying themes reflecting participants’ beliefs and experiences with respect to eating a zero-carb diet, and their definitions of a zero-carb diet. We followed an inductive thematic analysis approach, which involves moving from the specific to the general, whereby instances of interest (e.g., recurring pieces of conceptually relevant information provided by participants) are observed and synthesized into larger wholes or general statements (i.e., codes, subthemes, and themes). Our a priori decision was to code responses to the point of theoretical saturation and to include at least 30% of participants’ responses (*n* = 51) to ensure adequate coverage of participants’ beliefs. Similar cut-offs have been followed in other belief elicitation studies [34]. Across themes, theoretical saturation, the point at which no new concepts emerge, was observed well before coding the final participant’s responses, but we continued coding to ensure saturation was achieved. We conducted a separate thematic analysis on every question asked, resulting in different sample sizes for the themes emerging from each question. Our unit of analysis was a single response, a complete participant quotation.

ThematiCoder v.1.0 [35] was used to organize the thematic analyses. ThematiCoder 1.0 is a software application that enables the recording and coding of participant quotations and themes, while automatically calculating and displaying useful statistics, such as number, percentage, and coverage of quotes within and across codes or themes. The ThematiCoder v.1.0 spreadsheets used in the current study are available in the Appendix A.

Thematic analyses were conducted by the first author and reviewed against Braun and Clarke’s [33] 15 criteria for a high-quality thematic analysis conducted by the second and third authors. These criteria appraise aspects of all stages of thematic analyses (e.g., thoroughness, inclusiveness, and comprehensiveness of codes; coherence and distinctiveness of themes; interpretation of data instead of the description of data; representativeness of quotations; consistency between stated methods and reported findings; transparency of the written report). All quality criteria were met in the present study. In addition, the authors engaged in reflexive analysis throughout the course of the research, following Willig [36].

## 3. Results

### 3.1. Participant Characteristics

Participants comprised 170 adults (M = 42.82 years of age, SD = 12.05) from 25 countries. There was no evidence of disordered eating in the sample, and only a small number of participants (*n* = 15, 8.82%) reported being diagnosed with a mental health condition at the time of data collection. The completion rate, that is, the extent to which participants completed the questionnaire, was very good (78%). Technical issues relating to WIFI/internet connectivity and computer malfunction were reported as reasons for non-completion. All participants with one exception (*n* = 169, 99.41%) reported intentions to continue the diet indefinitely. Detailed participant information is presented in Table 1.

### 3.2. Thematic Analyses

Thematic analyses were structured under two overarching categories: (1) zero-carb diet definition and (2) lived experiences with a zero-carb diet. Each category comprised a set of themes.

#### 3.2.1. Category 1: Zero-Carb Diet Definition

Participants defined their diets in terms of types of foods consumed but also in terms of daily, weekly, and even yearly routines relating to the diet. We describe participants’ conceptualization of their diet under three themes: meat and water first; idiosyncratic eating; and slowly meeting meat. Themes and illustrative quotations describing the zero-carb diet are presented in Table 2.

##### Theme: Meat and Water First

Overwhelmingly, participants ate a wide variety of animal-sourced foods for every meal, particularly meats and organs (e.g., beef, pork, poultry, sheep, organ meats, game); eggs; dairy products; fish and seafood. The most popular food choices were beef cuts (98.12% coverage across quotations); followed by fatty fish (72.12%); pork cuts (64.11%); eggs (62.65%); dairy products (55.94%); poultry (51.28%); organ meats (40.29%); animal fats (39.03%); cold meat cuts (26.81%); and shellfish (17.36%). Spices were used in cooking (37.93% coverage across quotations), with salt most frequently mentioned. Foods were also eaten raw (19.22% coverage across quotations). Dietary supplements were generally not popular (7.56%), but some participants mentioned taking vitamins, minerals, collagen powder, and organ meats in capsule form. Water—including mineral and carbonated water—was the beverage of choice (42.73% coverage across quotations), followed by coffee (39.08%); alcoholic drinks (18.44%); tea (15.14%); and sugar-free soft drinks (3.28%) (Table 2: P1, female, aged 55, USA).

Non-animal sourced foods with carbohydrate content were also eaten on occasion, including fruits (11.33% coverage across quotations), dark chocolate (7.74% coverage across quotations), honey (6.03% coverage across quotations), and vegetables (5.41% coverage across quotations). Deviations from animal-sourced foods were, however, rare, and typically occurred in social situations where participants felt pressured to eat what others were eating. Some participants deliberately scheduled eating foods with higher carb content, describing these instances as “cheats” or “treats” from their usual diets (Table 2: P1, female, aged 55, USA; P2, female, aged 49, UK).

##### Theme: Idiosyncratic Eating

Participants’ eating patterns were idiosyncratic. Many did not have fixed eating time schedules, mentioning that they would eat when hungry and “to satiety”. Related, participants explained that their chosen foods kept them sated for long periods of time, and as a result, they ate once or twice a day. Some participants described this eating pattern as intermittent fasting, one-meal-a-day or OMAD or, simply, fasting (29.12% coverage across quotations). Other participants mentioned having breakfast, lunch, brunch, and dinner, but those terms were not used in the traditional sense, as the same types of foods were consumed regardless of the time of day. Also, some participants snacked, but snacks were the same foods eaten during meals (Table 2: P3, male, aged 34, Germany; P4, male, aged 42, New Zealand).

##### Theme: Slowly “Meeting” Meat

For many participants, the decision to uptake a zero-carb diet was the culmination of a slow and gradual process, where carbohydrate consumption was steadily reduced. Participants typically ended up eating a meat-heavy diet, after having spent a substantial amount of time, sometimes years, eating ketogenic-type diets. Progressively excluding carbs while increasing meat and other animal food content was experienced as a natural, logical, and considered process. This was because the diet was adjusted based on conducting personal research on, and experimentation with, various foods and noting the physical and emotional responses to the foods (Table 2: P5, male, aged 32, Greece).

#### 3.2.2. Category 2: Lived Experiences with a Zero-Carb Diet

Participants’ experiences with the diet are reported under three themes: wellbeing and quality of life, challenges, and the internet as the site of social support and knowledge. Themes and illustrative quotations from participants’ lived experiences with the diet are presented in Table 3.

##### Theme: Wellbeing and Quality of Life

All participants reported improved wellbeing and quality of life since taking up the diet. These improvements are related to health; interpersonal relationships; learning and growth; and simplicity and independence. Specifically, a number of physical and psychological health benefits were attributed to the diet. Physical health benefits related to the digestive system (e.g., reduction or elimination of irritable bowel syndrome symptoms); integumentary system (e.g., smoother skin, stronger hair, and nails); and musculoskeletal system (e.g., reduced joint and back pain). Furthermore, participants experienced an improvement in metabolic health (e.g., blood pressure, glucose, insulin, and thyroid function, as evidenced by biomarkers) and sex drive. Psychological health benefits included improvements in mood and cognition, including a reduction in anxiety, depression, and tiredness; increased happiness and joy, confidence, self-esteem, mental clarity, memory function, sleep quality, and productivity at work. Participants also reported the experience of the pleasure arising from eating the foods (e.g., the taste and texture were inherently pleasurable). Furthermore, some participants mentioned that the diet facilitated participation in health-promoting behaviors, specifically exercise and fasting. During exercise, participants felt stronger and more energized, but also, experienced the benefits of exercise more pronounced (e.g., more muscle-building) (Table 3: P6, male, aged 58, South Africa; P7, male, aged 28, USA).

Preoccupation with the diet was also reported to foster the development of relationships with like-minded people. Participants sought out, interacted with, and formed zero-carb networks, some as in-person groups, but mostly through online platforms and social media. The networks included a diverse international community of people from multiple educational and occupational backgrounds. These relationships were important sources of acknowledgment, approval, and support, as well as motivation for study, research, and personal experimentation with the diet. In the absence of published research on the diet and the divergence from mainstream food recommendations, participants relied heavily on their networks to receive and share information. The diet was often tweaked, as a function of continued study, self-experimentation with foods, and information-sharing with the networks. However, participants’ relationships and daily interactions with their immediate social circles (e.g., spouse, close friends, close family members, co-workers) were also important sources of support and influence in terms of adhering to the diet (Table 3: P8, female, aged 35, Australia).

A dimension of the diet contributing to participants’ quality of life was the simplicity of the diet. Participants explained that the diet saved them time in shopping and cooking the foods, but also money. The minimalistic nature of the diet seemed particularly suitable to those who were autonomous and independent in terms of character, living, and working conditions, and those familiar with seeking out, using, and assimilating health-related information on their own. The diet seemed to attract independent people but also to strengthen existing independent tendencies (Table 3: P9, female, aged 51, Canada; P10, male, 36, The Netherlands).

##### Theme: Challenges

In parallel with improvements in wellbeing and quality of life, participants were challenged in their pursuit of excluding carbohydrates and prioritizing animal-sourced foods. These included interpersonal, structural, and internal challenges.

Disclosing the nature of their diet and the larger goal of excluding carbohydrate intake altogether were identified as sources of strain on interpersonal relationships. Participants understood that their chosen eating style was somewhat socially unacceptable and isolating and tended to elicit stigmatizing responses from others. The term social stigma was used by participants to describe reactions they commonly received from others around them that followed conventional eating patterns. These “others” also included healthcare professionals. Participants felt “tired” or “fed up” having to explain their eating preferences to others, especially since their explanations would often lead to heated debates and arguments over diet. Furthermore, in social situations, participants were often pressurized to eat foods that did not fit with their diet such as foods with high sugar or carbohydrate content. Participants believed that almost everyone disapproved of their chosen way of eating (Table 3: P11, male, aged 27, UK).

Participants also experienced structural challenges to following the diet, that is, societal, financial, infrastructural, and policy-related obstacles. Participants understood that society or the environment was generally structured to facilitate consumption of foods high in sugar and carbohydrates, and, consequently, made consumption of zero-carb foods particularly difficult. These structural barriers reflected broad, interlinked ideological and educational systems, policies and industries, and infrastructures. For example, participants argued that medical and nutritional education prepared physicians and dieticians, as well as the organizations that these health experts belonged to, to strongly disapprove of diets heavy in animal-sourced foods. Instead, health-related systems collectively promoted foods higher in sugars and carbohydrates.

Participants also observed how the food industry tended to produce and advertise foods high in sugar and carbohydrates. Similarly, they also identified the restaurant and wider hospitality industry, as well as supermarkets and grocery stores, as posing potential obstacles to the diet, given the heavy emphasis on high-carbohydrate foods. Eating out of the home (e.g., in restaurants, while traveling, at work, on business meetings) was identified as particularly challenging due to limited provision and the high cost of animal-sourced foods. It was also mentioned that many organizations and groups popularize ideologies that portray meat and animal-sourced diets as harmful. These organized groups including vegetarians, vegans, animal rights advocates, and environmentalists, were perceived as presenting antagonistic views that challenged their views on diet. Furthermore, participants understood that social media, advertising, and film industries promoted diets high in carbohydrates and opposed diets based on animal foods (Table 3: P12 male, aged 60, USA; P13, female, aged 49, USA).

Finally, participants experienced internal challenges relating to the diet. For example, some experienced physiological discomfort when trying a zero-carb diet for the first time often described as “the adaptation phase” or “keto flu”. Similar symptoms were also experienced when deviating from the diet once past the adaptation phase. This discomfort was typically transient, and related to gastrointestinal reactions, but also to “immunological” and “psychological” reactions. The diet also resulted in some participants being “more sensitive” or “less tolerant” to eating foods high in carbohydrate content. Other participants noted feeling bored with the limited food options available to them, as well as uncertain about the long-term consequences of the diet given the lack of published research on it (Table 3: P14 male, aged 34, USA; P15, female, aged 40, UK).

##### Theme: The Internet as the Site of Social Support and Knowledge

Social interactions through various online platforms emerged as a pervasive influence on participants’ diets. Nearly all participants were influenced to uptake the diet by an online advocate of the diet. High-profile zero-carb diet advocates vocal on social media were a primary source of information, influence, and support. These advocates included healthcare experts, including physicians, academics, researchers, and dietitians but also people with very little or no health-related training or expertise. Participants listened to podcasts, watched YouTube clips, and read websites and blogs that were dedicated to the diet. Many participants were members of zero-carb or “carnivore” communities on social media platforms. Membership of these online communities facilitated online interactions between participants and like-minded people who also followed the diet. A sense of belonging to an online community, or online family, was an important aspect of participants’ lives. The internet was also the primary space for obtaining and sharing knowledge on zero-carb diets. For some participants, these online communities were the only source of information and support in relation to their diet, given the limited acceptance of the diet in real-life social circles and healthcare professionals. Belonging to these online zero-carb communities, expressing and sharing values, beliefs, and practices suggests an emergent zero-carb group identity (Table 3: P16, male, aged 33, UK; P17, female, aged 51, Canada; P18, female, aged 37, UK).

## 4. Discussion

The present study explored the beliefs and experiences of people eating a zero-carb diet for a minimum of six months using an online qualitative survey method distributed via Twitter, a social media platform. Participants were asked to describe and elaborate on the advantages, disadvantages, normative influences, facilitators, and detractors of eating a zero-carb diet. Data were thematically analyzed. Themes presented participants’ shared understanding and definition of the diet, as well as their experiences with the diet.

### 4.1. Zero-Carb Diet: Definition and Wellbeing

A basic definition of a zero-carb diet emerged based on participants’ shared dietary habits. According to participants, a zero-carb diet involves the consumption of a variety of foods that are predominately, but not exclusively, animal-sourced. The types and amounts of foods consumed, as well as the timing of the meals, were idiosyncratic. “One”, singular, zero-carb diet did not emerge, and therefore, the term “zero-carb diet” does not refer to a specific, clearly-bound set of foods, macronutrient ratios, or the complete elimination of carbohydrates. While the label zero-carb diet is empirically imprecise, it is an appropriate descriptor for the diet because it captures participants’ shared understanding of the development and evolution of their dietary practices, their goal to reduce carbohydrate intake to a bare minimum, and the psychophysical changes resulting from carb restriction/exclusion. For example, many participants in the current study described a process of gradual carbohydrate reduction, originating from paleo, low-carb, or ketogenic eating styles. This suggests a continuum of carbohydrate restrictive diets, with zero-carb variants located at the extreme end of the continuum. Participants reported that they arrived at a zero-carb variant that suited them and gave them the most health benefits, through personal study, food experimentation, and interaction with like-minded people. In the present study participants also reported experiencing a wide variety of wellbeing benefits that they attributed to their diet, including better digestion, better metabolic health biomarkers, better physique, vitality, reduced hunger, improved mood, joy, knowledge-gaining, personal development and experimentation, and social relationship development. Furthermore, a notable finding from the current data is that a zero-carb diet also seemed to covary or facilitate other health-promoting strategies, including fasting and exercising. Taken together, the data from the current research provide important information on the label of this particular variant of a low-carbohydrate diet, the types of expected eating patterns shared by those following the diet, and insight into the ways individuals adopt and refine their dietary practices while following the diet.

### 4.2. Challenges

The zero-carb diet introduced challenges to participants’ lives. Some of the challenges were relatively innocuous daily hassles, such as eating out and grocery shopping challenges, and boredom due to limited food types consumed. Other challenges were more serious, with potentially negative health implications, such as experiencing or expecting to experience stigma, strife in interpersonal relationships, lack of guidance and support from healthcare providers, adverse physiological symptoms, and worry about the long-term healthfulness of the diet. Nevertheless, participants framed challenges as transient or not significant enough to derail maintaining their diet. If anything, overcoming these challenges appeared to bolster participants’ resolve in adhering to the diet. All but one participant reported intentions to follow the diet indefinitely, and most reported that others’ disapproval of the diet did not matter enough to stop them. Participants largely overcame challenges through personal research and experimentation with foods and closely interacting with online communities, often described as families. These findings are congruent with other research on individuals following carbohydrate-restrictive diets, where participants reported a strong desire to continue with the diet, despite similar challenges [37,38]. It appears that a strong group identity based on adherence to so-called unconventional diets helps to override or negate challenges, through bonding among group members over shared values, goals, and outcomes of the diet. Comparable experiences have been reported by individuals following vegetarian and vegan diets, with a substantive body of research documenting strong, cohesive social identities among individuals identifying as vegetarian and vegan [39,40,41]. By contrast, research into the social identities of those following carbohydrate-restrictive diets is scarce, although there are some preliminary studies [37,42]. While the present study did not specifically set out to identify a zero-carb identity, a key emergent finding from our data suggests that such an identity appears to exist.

### 4.3. Implications for Research and Practice

Based on the present findings, we recommend additional research in several domains. First, we recommend research into the safety and efficacy of excluding or severely limiting carbohydrates for extended periods of time. Our data indicate that people eating zero-carb diets for at least six months experience important benefits but also shortcomings, all of which warrant further investigation, especially through clinical trials. In addition, it would be important to account for changes in beliefs, motives, and perspectives on zero-carb diets in the participants of these trials. This will provide important data on how the efficacy and experience of change affect individuals’ beliefs, such as those identified in the current study.

Based on the current data, we note that people eating zero-carb diets are unlikely to receive support and care from conventionally trained health professionals, who have not received specific training on these dietary practices or have insufficient background information on the health and social implications of carbohydrate restriction. We also note that people eating zero-carb diets experience stigma from those around them that are unfamiliar with this pattern of eating, including healthcare providers. Prior evidence overwhelmingly suggests that experience—and even the anticipation—of stigma is a major barrier to seeking and receiving healthcare and quality of life [43,44]. Consequently, participants in the present study sought and depended on sources of information and support outside their regular healthcare providers, typically from online zero-carb communities. While many in these communities may have appropriate backgrounds and knowledge of the diet, there is also a risk that people may receive information that lacks an evidence base or credibility. Lack of support and stigma by healthcare providers has also been documented among people following other types of carbohydrate-restrictive diets [38,45]. It is also documented that healthcare providers who support, or express intention to support, those on carbohydrate-restrictive diets experience stigma themselves by their colleagues [46], reflecting a relative lack of acceptance and knowledge among the mainstream healthcare community. Given participants’ expressed strong motives to adhere to the diet, with or without healthcare provider support, we suggest that healthcare providers seek to keep informed and upskill themselves to support this population. Related, we advocate research into how healthcare provider stigma can be reduced and how healthcare providers can support and monitor people on these diets.

We also recommend research into the formation of social identities emerging from carbohydrate restriction. It is well established that food choices are a means to express personal values, beliefs, and worldviews [37,39]. Our findings suggest that a zero-carb social identity exists. These findings are consistent with social identity theory [47], which suggests that people develop social identities based on the shared beliefs and attitudes of the social groups to which they belong (ingroups) and that such identities are a source of motivation and self-esteem. Our participants shared values, beliefs, and practices; belonged to groups based on their food choices and viewed these ingroups favorably. Although carbohydrate-restrictive diets have a long history, with formalized (medical) supervised plans appearing in the early 1920s [48], the shared identity of people following these types of diets has been largely overlooked, but limited data exist [37]. Future intervention research, therefore, may seek to capitalize on such social identities as a means to promote adoption and maintenance of zero-carb diets among those interested in, and receptive to, adopting a zero-carb diet. Numerous behavior change strategies have that tap into such identities have been identified and these may inform the development of the content of such interventions [49] (pp. 649–660, 225–236).

Finally, we recommend research into how people following zero-carb diets, and other carbohydrate-restrictive diets, maintain their diets in face of challenges presented in the modern dietary landscape. One of the consistent barriers discussed by participants in the current study was the high availability of high-carb foods and beverages, and the relative lack of low-carb or zero-carb alternatives. Participants, like most people in Western nations, are consistently being marketed foods high in carbohydrates and sugars. Participants felt acutely that they were going against societal norms in their efforts to exclude carbohydrates, experiencing derision and stigma as a result. Exploring how those on zero-carb, and other carbohydrate-restrictive diets, navigate through and resist environments where there is high availability of high-carbohydrate and high-sugar foods, can inform the development of strategies promoting consumption of alternatives to such foods. Given the well-documented link between high carbohydrate consumption and metabolic health conditions like type 2 diabetes and obesity, such an approach is likely to have broad implications for health.

### 4.4. Strengths and Limitations

The present study was the first to explore beliefs and experiences of people eating zero-carb diets and to elicit a definition of a zero-carb diet, filling an evidence gap. Other strengths of the study included the adoption of fit-for-purpose and novel qualitative methods and recruitment of a large, international sample of people who had been voluntarily on a zero-carb diet for at least six months, with a high completion rate. The use of the online qualitative survey method afforded additional advantages including a “wide-angle lens” approach to encompass diverse views and experiences of participants; the elicitation of “within-group” views from hard-to-reach populations; heterogeneity in views by covering a larger cross-section of the population of interest; participant anonymity and control over the location and time of participation by participants; and the minimization of interviewer demand characteristics. Furthermore, our procedure has clear reproducible steps allowing others to replicate the approach.

In addition, the study was developed based on established study quality criteria [27], extant frameworks of online qualitative survey methodology [23,24], and analyses were appraised against Braun and Clarke’s [33] criteria for high-quality thematic analysis. Finally, for the purposes of maintaining trustworthiness and transparency of our data, we stated the first author’s personal experimentation with a zero-carb diet and provided all study materials, data files, and data analyses in the Appendix A.

These strengths notwithstanding, current findings should be interpreted in light of some limitations. A potential limitation is participants’ self-selection bias. Participants volunteered their participation and were likely those with high involvement and immersion in the diet and high motivation to continue with the diet. Consequently, their views may differ from others on a zero-carb diet who did not elect to participate, and those who were not reached through our recruitment strategy. A related limitation is that our recruitment and sampling method did not allow us to estimate response rates, and, therefore, we could not ascertain refusal rates, which may have provided an indication of the extent of self-selection. Although a random stratified sample is highly desirable as it enables better generalization of findings to the broader population, the current sample comprised participants with highly-specialized interests and behaviors for which no clear norms exist making it challenging to recruit a sample that is representative of all followers of low-carb dieters. In addition, although we recruited a large sample from multiple backgrounds and national groups, most participants were of high socioeconomic status (Table 1). This places limits on the extent to which current findings can be generalized to all individuals following zero-carb diets. Future research should consider pro-active recruitment strategies that reach zero-carb dieters who do not engage with social media and those who do not tend to respond to unsolicited requests for participation.

### 4.5. Reflexive Analysis

We aimed to collect and analyze and present participants’ perspectives and experiences with a zero-carb diet fairly and disinterestedly, but we acknowledge that we, as researchers play a role in constructing knowledge, so we were mindful to be reflexive throughout the course of the research. The first author is an applied health psychologist with an interest in human nutrition, experience in mixed-methods psychological research, and extended experience with eating a zero-carb diet. The second author is an applied biological scientist, with expertise in interdisciplinary research relating to human and animal health and wellbeing; they have no experience with a zero-carb diet. The third author is an applied health and social psychologist with experience in mixed-methods psychological research; they have no experience with a zero-carb diet. The research we have produced likely benefits from pooling expertise from different academic backgrounds; an insider approach to a zero-carb diet, and an approach based on an empirical understanding of zero-carb eating.

## 5. Conclusions

The current study contributes to knowledge by providing the first analysis of beliefs and experiences of individuals on a zero-carb diet, and their shared understanding of the diet. The research suggests that participants define a zero-carb diet as one that involves the severe restriction of carbohydrate consumption and emphasizes animal-sourced foods. Participants cited numerous health and well-being advantages as key reasons for taking up and maintaining the diet. Participants also reported strong intentions to continue with the diet indefinitely, despite lack of published evidence on the diet, lack of support by healthcare professionals and significant others, and stigmas attached to the diet. Furthermore, responses indicate the emergence of a zero-carb social identity, facilitated by the membership of online zero-carb communities. Based on these findings, we recommend further research into the long-term efficacy and safety of the diet, education initiatives to inform healthcare providers of the rationale and reasons why individuals choose to follow these diets, and further investigation of ways that healthcare providers can support those wishing to uptake, maintain, or discontinue the diet. We also recommend research into the social identities of people on carb-restrictive diets, the ways these social identities shape initiation and adherence to the diets, and how individuals can help dispel myths and prejudice surrounding these diets.

## Figures and Tables

**Table 1 behavsci-11-00161-t001:** Sample Characteristics.

Characteristic	Value	Characteristic	Value
Age *M*	42.82	Occupation	
*SD*	12.06	Managerial/Marketing/Sales	25 (14.70)
Gender		Healthcare provider	18 (10.58)
Male	111 (65.29)	IT and programming	16 (9.41)
Female	58 (34.12)	Teacher/Professor	16 (9.41)
Other	1 (0.59)	Engineer	12 (7.05)
Country of residence		Technician	11 (6.47)
USA	87 (51.17)	Artist	9 (5.29)
UK	22 (12.94)	Administration	9 (5.29)
Canada	13 (7.64)	Self-employed	9 (5.29)
Australia	10 (5.9)	Services provider	7 (4.11)
South Africa	6 (3.52)	Business/Finance	9 (5.29)
Netherlands	5 (2.94)	Retired and homemakers	6 (3.52)
Germany	3 (1.76)	Physicians	4 (2.35)
New Zealand	3 (1.76)	Attorneys/Solicitors	4 (2.35)
Denmark	2 (1.17)	Civilservants/Gov. employees	4 (2.35)
Finland	2 (1.17)	Languages	2 (1.17)
Greece	2 (1.17)	Consultants	2 (1.17)
Sweden	2 (1.17)	Tourism	2 (1.17)
Belgium	1 (0.58)	Unemployed	2 (1.17)
Brazil	1 (0.58)	Agriculture	1 (0.58)
Estonia	1 (0.58)	Statistician	1 (0.58)
Japan	1 (0.58)	Student	1 (0.58)
Mexico	1 (0.58)	Currently have mental health condition	
Norway	1 (0.58)	Yes	15 (8.82)
Russia	1 (0.58)	No	155 (91.18)
South Korea	1 (0.58)	ESP	
Spain	1 (0.58)	ESP item 1	169 (99.41)
Switzerland	1 (0.58)	ESP item 2	13 (7.65)
Taiwan	1 (0.58)	ESP item 3	115 (67.65)
Turkey	1 (0.58)	ESP item 4	24 (14.12)
UAE	1 (0.58)	Intention to continue a zero-carb diet	
Education level		Yes	169 (99.41)
University	123 (72.35)	No	1 (0.59)
Non-university HE	24 (14.12)		
High-school	18 (10.59)		
Elementary school	1 (0.59)		
Other	4 (2.35)		

Note. Values represent numbers of participants with percentage in parentheses unless otherwise specified. ESP = Eating Disorder Screen for Primary Care.

**Table 2 behavsci-11-00161-t002:** Zero-Carb Diet Definition.

Themes	Illustrative Quotes
Meat first	“Beef rib eye steak every day…Occasionally a few seared shrimps, scallops or oysters with it. Raw beef liver weekly. Sushi twice a month or so including salmon, yellow tail, squid. Biltong made from beef chuck and salt for travel or lunch at work. Water to drink. Very occasional fizzy mineral water but mostly flat well water. Salt. Bone broth every evening home made with a pressure cooker from beef feet and home grown chicken. Served with salt and a tablespoon of collagen powder. Oysters on the half shell about once a month” (P1, female, aged 55, USA).
	“…Rarely, I eat strawberries, raspberries and blueberries. I occasionally eat a couple of squares of dark chocolate as a treat…” (P2, female, aged 49, UK).
Idiosyncratic eating	“I eat one-meal-a-day (OMAD) at least 4 days per week. Only eating when hungry and sometimes fasting for 48 h” (P3, male, aged 34, Germany).
	“Breakfast: Beef liver (about 300 g), grilled with butter; glass of cold-brew coffee. Lunch: Beef sirloin steak (about 300 g), fried in butter; glass of water. Dinner: Beef sirloin steak (about 300 g), fried in butter; glass of water” (P4, male, aged 42, New Zealand).
Slowly meeting meat	“The first 2 years were mostly revolving around the so called “classical bodybuilding” diet…I slowly veered away from lean meats and went…towards a ketogenic diet. Since February 2019, I have been on a Carnivore Diet, removing ALL fiber...” (P5, male, aged 32, Greece).

Note. *p* = Participant.

**Table 3 behavsci-11-00161-t003:** Lived Experiences with a Zero-Carb Diet.

Theme	Illustrative Quotes
Wellbeing	“My overall health has drastically improved. I had constant stomach issues eating even small quantities of vegetable and these have basically gone after going zero-carb. Depression and brain fog have gone and probably for these reasons alone I will not change the way of eating” (P6, male, aged 58, South Africa).
	“Best physique of my life, despite spending less time lifting weights daily than I had for the previous 8 years (and without any steroids or supplementation)—strongest I’ve ever been in my life even with less time in the gym-lowest body fat I’ve ever been without even actively trying to lose weight/fat” (P7, male, aged 28, USA).
	“I developed connection with like-minded individuals in the zero-carb community. Opened my eyes to what ‘healthy’ really can be. Gained an increase in knowledge of health, wellness, diet and lifestyle (and I am constantly learning and evolving, daily)” (P8, female, aged 35, Australia).
	“I am a widow and live alone so no one else to cook for or eat with most of the time I’m 51 and care less about what others think. The meat I eat is easily accessible to buy and cook and financially affordable” (P9, female, aged 51, Canada).
	“I have a low need for social validation and interaction-Financially well off enough to make it affordable-Enough intellectual and education background to trust my own research” (P10, male, 36, The Netherlands).
Challenges	“It is socially quite paralysing, unless you make an effort to find like-minded individuals with whom you could dine…As you are deviating from societal norms, you attract lots of derision from friends and family-You don’t feel like you belong quite as much on sociable occasions when you choose to reject almost all food and drink on offer. Socially people are always trying to ‘talk me into’ carbs. ‘One bite won’t hurt you’. ‘It’s just a piece of cake, it won’t make or break you’” (P11, male, aged 27, UK).
	“Any and all doctors, medical experts or nutritionists solely advocate a vegetable/carb-based diet” (P12, male, aged 60, USA).
	“It is difficult to order ‘meat only’ meals sometimes as people often get the order wrong…You pay the same price for just a meat only meal plate as a meat/side/bun/dessert meal plate” (P13, female, aged 49, USA).
	“I have less tolerance for non-ZC foods. Initially, diarrhea (1–2 weeks every other day), then normalized” (P14 male, aged 34, USA).
	“Options are severely limited. I occasionally get bored…” (P15, female, aged 40, UK).
	“The diet is not backed by a lot of scientific studies or research material, so it could be damaging in the long term” (P16, male, aged 33, UK).
Internet families	“The support that I have found online has been paramount to my staying the course and adjusting my eating habits over time” (P17, female, aged 51, Canada).
	“My Twitter ‘family’ of LCHF/Carnivore plays an important role in providing indirect support through information and general sharing of experiences” (P18, female, aged 37, UK).

Note. Illustrative quotes taken from open-ended survey questions on behavioral beliefs. Note. *p* = Participant.

## Data Availability

Study materials and data that support this study are available online: https://osf.io/htxpv (Appendix A).

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
