# Peer review of "Beliefs and Experiences of Individuals Following a Zero-Carb Diet"

_behavsci, 2021, doi:10.3390/bs11120161_

Round 1

Reviewer 1 Report

The authors explain very well the results obtained on the online survey about Zero-Carb Diet and are able to organize the results in a clear way to understand the subject and the comments from the Zero-Carb dieters.

However, the manuscript lacks important information on the aspect of health results, for example, weight amount before starting the diet and current weight when answering the online survey. Do they also exercise while on the diet? Have any of the participants have had a doctor's appointment to verify their health, specifically their weight, blood pressure, etc.? 

In addition, the population is mostly diverse from different countries, with only one person answering the survey, a specific population should be selected. The motives are not very clear on why they are doing this specific diet, other than what they read online. Have they done other diets before and what are the comparisons?

Author Response

REVIEWER’S COMMENT: The authors explain very well the results obtained on the online survey about Zero-Carb Diet and are able to organize the results in a clear way to understand the subject and the comments from the Zero-Carb dieters.

AUTHORS’ RESPONSE: We thank the Reviewer for their positive comment in relation to the organization and explanation of our results.

REVIEWER’S COMMENT: However, the manuscript lacks important information on the aspect of health results, for example, weight amount before starting the diet and current weight when answering the online survey. Do they also exercise while on the diet? Have any of the participants have had a doctor's appointment to verify their health, specifically their weight, blood pressure, etc.? 

AUTHORS’ RESPONSE: We thank the Reviewer for raising these issues. We agree that obtaining comparative (before-during-after) data on zero-carb dieters’ health measures and outcomes would be an informative study. However, the purpose of the present research was to explore participants’ beliefs and experiences of zero-carb diets, particularly the beliefs that underpin their decisions to begin and maintain a zero-carb diet. Measurement and comparison of health outcomes was not part of the present research, and it would be difficult to make meaningful comparisons between such outcomes alongside the beliefs and motives reported using these qualitative data.

REVIEWER’S COMMENT: In addition, the population is mostly diverse from different countries, with only one person answering the survey, a specific population should be selected. The motives are not very clear on why they are doing this specific diet, other than what they read online. Have they done other diets before and what are the comparisons?

AUTHORS’ RESPONSE: We thank the Reviewer for raising important issues. Obtaining a diverse, international sample was intentional on our part and a strength of the present research in order to obtain participants’ demographical and diet particulars. Participants’ motives (i.e., reasons) for engaging the diet are provided in the results, in particular, the themes “wellbeing and quality of life” and “the internet as the site of social support and knowledge”. In addition, while obtaining information on participants’ previous dietary practices was not part of the aims of the present research, participants did volunteer information on their previous eating habits, especially their progression to a zero-carb diet from a low carbohydrate diet, as indicated in the theme “slowly ‘meeting’ meat”. Please see, for example, our sections on page 8, paragraphs 2 and 4, and on page 11, final paragraph and page 12, first paragraph of the manuscript.

Reviewer 2 Report

The topic and purpose of the manuscript are very important and novel. Overall, I consider the article a valuable piece of work. The authors have elaborated on a particularly current topic. The perceived strengths associated with the manuscript are greater than any weaknesses. In the following, I list the strengths and weaknesses of the manuscript:

  1. Carbohydrate-reduced or zero-carb diets are popular diets today. Yet there is relatively little relevant research on the topic. The reason for the choice of topic by the authors corresponds to reality.
  2. The perceived benefits of a carb-free diet certainly outweigh the perceived disadvantages. The opinions of experts are also quite divisive on the topic. It would be worthwhile to explain the perceived advantages and disadvantages even more thoroughly in the introduction part of the article. The authors have already made references to this, this could further prove the relevance of their research.
  3. I recommend to the authors an article that systematizes the perceived benefits and barriers of plant-based nutrition. This is somewhat the other extreme compared to the diet studied by the authors. However, there are common points so it can even be used as a reference or can be included in the introduction section. Source: https://www.mdpi.com/2071-1050/12/10/4136
  4. The qualitative research methodology used by the authors can be considered relatively unique. In conducting the research, the authors also used a questionnaire survey, which in many cases is a typical quantitative methodology. The authors, of course, minimized quantifiable questions and created mostly open-ended questions. My suggestion is that the methodology used should be explained in even more detail. Furthermore, it should be emphasized that the basic quantitative nature of the research results with qualitative values in the present research. In my own judgment, this is a mixed qualitative and quantitative method. This should be explained more thoroughly, even with more literature, based on the nature of the research.
  5. The authors write about further possible directions of research. I suggest that they should explain in more detail in a few sentences what specific further research can be built on the current survey. Furthermore, it would be worthwhile to specify that this other type of diet can be tested using a similar methodology.
  6. It is confusing for the reader that the link to the OSF platform (https://osf.io/htxpv/?view_only=be8c65de55dd472ca174e4cf0e7315e0) will be included three times in the short text of the article in its full version. It would be worthwhile to make a literature reference from this or put it in a footnote at the bottom of the page.

    All in all, a valuable article has been written, the publication of which I support following the minor additions listed above.

Author Response

REVIEWER’S COMMENT: The topic and purpose of the manuscript are very important and novel. Overall, I consider the article a valuable piece of work. The authors have elaborated on a particularly current topic. The perceived strengths associated with the manuscript are greater than any weaknesses. In the following, I list the strengths and weaknesses of the manuscript:

AUTHORS’ RESPONSE: We appreciate the Reviewer’s overall positive appraisal of our research and its value, and thank them for their useful suggestions to improve the manuscript. Please see our responses below, and our revisions marked in blue font in the revised manuscript.

  1. REVIEWER’S COMMENT: Carbohydrate-reduced or zero-carb diets are popular diets today. Yet there is relatively little relevant research on the topic. The reason for the choice of topic by the authors corresponds to reality.

AUTHORS’ RESPONSE: We thank the Reviewer for their positive comment in relation to the need of the present study, given the lack of research on the topic. Our goal was to fill an evidence gap on a topic and population that has, to date, received little research attention.

  1. REVIEWER’S COMMENT: The perceived benefits of a carb-free diet certainly outweigh the perceived disadvantages. The opinions of experts are also quite divisive on the topic. It would be worthwhile to explain the perceived advantages and disadvantages even more thoroughly in the introduction part of the article. The authors have already made references to this, this could further prove the relevance of their research.

AUTHORS’ RESPONSE: We appreciate the Reviewer for suggesting adding further information on the health benefits and risks of carbohydrate-restrictive diets. We have added relevant text in the Introduction to enhance the debate over carbohydrate-restrictive diets (please see page 2, paragraph 2 of the revised manuscript).

  1. REVIEWER’S COMMENT: I recommend to the authors an article that systematizes the perceived benefits and barriers of plant-based nutrition. This is somewhat the other extreme compared to the diet studied by the authors. However, there are common points so it can even be used as a reference or can be included in the introduction section. Source: https://www.mdpi.com/2071-1050/12/10/4136

AUTHORS’ RESPONSE: We thank the Reviewer for alerting us to this article, which we have now cited in the Discussion section (please see page 13, paragraph 1 of the revised manuscript).

  1. REVIEWER’S COMMENT: The qualitative research methodology used by the authors can be considered relatively unique. In conducting the research, the authors also used a questionnaire survey, which in many cases is a typical quantitative methodology. The authors, of course, minimized quantifiable questions and created mostly open-ended questions. My suggestion is that the methodology used should be explained in even more detail. Furthermore, it should be emphasized that the basic quantitative nature of the research results with qualitative values in the present research. In my own judgment, this is a mixed qualitative and quantitative method. This should be explained more thoroughly, even with more literature, based on the nature of the research.

AUTHORS’ RESPONSE: We agree with the Reviewer that the methods used in our article were ‘relatively unique’. It aligns with an emergent approach to qualitative data collection and analysis using online surveys, which has been advocated by leading qualitative theorists/researchers (e.g., Braun, Clarke, & Gray, 2017; Braun, Clarke, Boulton, Davey, & McEvoy, 2020). An explanation of online surveys as a qualitative research tool in its own right is provided by Braun et al. (Braun, V., Clarke, V., Boulton, E., Davey, L., & McEvoy, C. (2020). The online survey as a qualitative research tool. International Journal of Social Research Methodology, 1-14. https://doi.org/10.1080/13645579.2020.1805550). Our study uses one approach cited – the qualitative survey – which also included closed-ended questions to obtain demographic data. We nevertheless added a brief sentence to the description of the method to the manuscript to clarify that our focus was on demographically dispersed populations (please see page 3, paragraph 1 of the revised manuscript). We do not think that our study should be characterized as a mixed-methods study. Such a study would likely involve the collection, analysis, interpretation, and integration of both quantitative (e.g., from experiments, questionnaires with closed-ended questions) and qualitative data (e.g., from focus groups, interviews) in a single study or in a series of studies that investigate the same underlying phenomenon. To be described as mixed-methods, our study would also have had to investigate zero-carb dieters’ beliefs and experiences through, say, quantitative questionnaires and interviews.

  1. REVIEWER’S COMMENT: The authors write about further possible directions of research. I suggest that they should explain in more detail in a few sentences what specific further research can be built on the current survey. Furthermore, it would be worthwhile to specify that this other type of diet can be tested using a similar methodology.

AUTHORS’ RESPONSE: Thanks for this suggestion, we have explained the specific additional future research emerging from these data in section 4.3 (please see page 13, paragraph 2 and page 14, paragraph 2 of the revised manuscript).

  1. REVIEWER’S COMMENT: It is confusing for the reader that the link to the OSF platform (https://osf.io/htxpv/?view_only=be8c65de55dd472ca174e4cf0e7315e0) will be included three times in the short text of the article in its full version. It would be worthwhile to make a literature reference from this or put it in a footnote at the bottom of the page.

AUTHORS’ RESPONSE: We agree with the Reviewer in that the OSF link that we originally included in the manuscript was very long and may have inhibited readability. However, the purpose of this link is to be consistent with blind peer review. However, we note now that this journal uses single-blind peer review anyway, so we have replaced it with the un-masked version which is much shorter: https://osf.io/htxpv.

REVIEWER’S COMMENT: All in all, a valuable article has been written, the publication of which I support following the minor additions listed above.

AUTHORS’ RESPONSE: We thank the Reviewer for their encouragement and constructive comments, which have helped improve our manuscript.
